# Corticotropin-Releasing Hormone: A Novel Stimulator of Somatolactin in Teleost Pituitary Cells

**DOI:** 10.3390/cells12242770

**Published:** 2023-12-05

**Authors:** Ruixin Du, Xuetao Shi, Feng Chen, Li Wang, Hongwei Liang, Guangfu Hu

**Affiliations:** 1Hubei Hongshan Laboratory, College of Fisheries, Huazhong Agricultural University, Wuhan 430070, China; duruixin@webmail.hzau.edu.cn (R.D.); shi_xuetao@ctg.com.cn (X.S.); 19838067154@163.com (F.C.); wli@webmail.hzau.edu.cn (L.W.); 2Key Lab of Freshwater Biodiversity Conservation Ministry of Agriculture, Yangtze River Fisheries Research Institute, The Chinese Academy of Fisheries Sciences, Wuhan 430223, China

**Keywords:** pituitary, neurointermediate lobe, stress, signal transduction, lipid metabolism

## Abstract

**Highlights:**

**What are the main findings?**
CRH could significantly induce pituitary somatolactin secretion and synthesis.CRH-induced SL expression was mainly mediated by the AC/cAMP/PKA pathway.Single-cell transcriptome showed that SLα and SLβ were expressed in different cells of the pituitary NIL region.SLα and SLβ could regulate hepatocyte lipid metabolism.

**What is the implication of the main finding?**
This study supplemented the function of CRH in the grass carp pituitary.This study suggested that there might be another pathway for the CRH response to stress.

**Abstract:**

Corticotropin-releasing hormone (CRH) is known for its crucial role in the stress response system, which could induce pituitary adrenocorticotropic hormone (ACTH) secretion to promote glucocorticoid release in the adrenal gland. However, little is known about other pituitary actions of CRH in teleosts. Somatolactin is a fish-specific hormone released from the neurointermediate lobe (NIL) of the posterior pituitary. A previous study has reported that ACTH was also located in the pituitary NIL region. Interestingly, our present study found that CRH could significantly induce two somatolactin isoforms’ (SLα and SLβ) secretion and synthesis in primary cultured grass carp pituitary cells. Pharmacological analysis further demonstrated that CRH-induced pituitary somatolactin expression was mediated by the AC/cAMP/PKA, PLC/IP3/PKC, and Ca^2+^/CaM/CaMK-II pathways. Finally, transcriptomic analysis showed that both SLα and SLβ should play an important role in the regulation of lipid metabolism in primary cultured hepatocytes. These results indicate that CRH is a novel stimulator of somatolactins in teleost pituitary cells, and somatolactins may participate in the stress response by regulating energy metabolism.

## 1. Introduction

Corticotropin-releasing hormone (CRH) is one member of the corticotropin-releasing hormone family, which consists of CRH, urocortin-1 (UCN1, called UTS1 in fish), urocortin-2 (UCN2), and urocortin-3 (UCN3) [1]. CRH was first isolated from the hypothalamus of sheep [2]. It is known for its crucial role in the stress response system. In mammals, CRH operates via the hypothalamic–pituitary–adrenal axis (HPA), where it is released from the hypothalamus into the pituitary gland, subsequently entering the anterior pituitary gland and binding to the CRH receptors (CRHR). This process leads to the production and release of adrenocorticotropic hormone (ACTH), which could stimulate the synthesis and release of glucocorticoids in the adrenal gland [3,4,5]. In teleosts, peptides of the CRH family were involved in the fish osmoregulation, glucocorticoid release (stress), food intake, and body exercise, through the hypothalamus–pituitary–interrenal (HPI) axis [6,7,8,9]. CRH could also promote the fish pituitary ACTH release and synthesis [10]; however, little is known about other pituitary actions of CRH in teleosts.

Somatolactin (SL), an important member of the growth hormone (GH)/prolactin (PRL) family, is a pituitary hormone unique to fish, which is released from the neurointermediate lobe (NIL) of the posterior pituitary [11]. Two isoforms of SL, SLα and SLβ, have been identified in the fish pituitary, e.g., in zebrafish [12], salmon [13], and grass carp [14], and their functions are suspected to have overlaps and differences [15]. In fish models, SL has been reported to be involved in multiple physiological processes, including chromatophore proliferation and differentiation [16], pigment aggregation [17], inflation of the swim bladder during embryo development [15], reproduction [13], lipid metabolism [18], and osmoregulation [19]. In addition, several studies have shown that SL was involved in stress responses. In rainbow trout (*Oncorhynchus mykiss*), environmental stress could cause the rapid activation of SL-secreting cells [20]. In female thin-lipped gray mullet (*Liza ramada*), several stress factors, such as handling, seawater acclimation, and confinement, could affect the pituitary SL secretion [21]. In addition, cadmium treatment could significantly induce the SLα expression in rare minnows [22]. These studies suggested that SL played an important role in the stress response. However, little is known about regulation mechanisms of SL in the stress responses.

In the present study, the correlation between CRH and somatolactins and their functions in stress were examined in grass carp (*Ctenopharyngodon idellus*), a commercial fish in Asian countries with a high market value [23]. As the first step, grass carp CRH and two somatolactins (SLα and SLβ) were cloned and their active proteins were synthesized. In addition, pharmacological methods were used to examine the possible involvement of AC/cAMP/PKA, PLC/IP3/PKC, and Ca^2+^/CaM/CaMK-II cascades in the regulatory actions of CRH on SLα and SLβ secretion and gene expression. Finally, using primary culture of grass carp hepatocytes as the model, we further examined the functions of somatolactins in teleost liver. This study proposes a novel mechanism by which SL participates in stress by responding to CRH.

## 2. Materials and Methods

### 2.1. Fish Acquisition, Acclimation, and Sacrifice

One-year-old grass carp (*Ctenopharyngodon idellus*) with a body weight of 1.0 to 1.5 kg and a body length of 40 to 50 cm were acquired from local markets and maintained for one week in well-aerated 250 L aquaria under a 14 h light/10 h dark photoperiod at 20 ± 2 °C. During the process, the fish were anesthetized in 0.05% MS222 (Sigma, St. Louis, MO, USA) followed by spinosectomy, according to the protocol approved by the committee for animal use at Huazhong Agricultural University (Ethical Approval No. HBAC20091138; date: 15 November 2009).

### 2.2. Chemical Reagents

Carp CRH1a and UTS1 were synthesized by GenScript (Nanjing, China), and the carboxyl-terminus of individual peptides was amidated. H89, MDL12330A, Dic-8, TPA, U73122, GF109203X, Nifedipine, KN62, and calmidazolium were obtained from Calbiochem (San Diego, CA, USA) (for background information on the test substances, please refer to Appendix A). These pharmacological agents were prepared as 10 mM frozen stocks in small aliquots and diluted with pre-warmed culture medium to appropriate concentrations 15 min prior to drug treatment.

### 2.3. Cloning and Tissue Distribution of Grass Carp CRHs and CRHRs

Total RNA was extracted from grass carp pituitary and reverse-transcribed with HifairTM III 1st Strand cDNA Synthesis Kit (Yeasen, Shanghai, China). Meanwhile, specific primers designed for the putative sequences were used to clone the whole target genes of CRHs, UTS1, and CRHRs. These gene sequences are from the National Center for Gene Research, Chinese Academy of Sciences (http://www.ncgr.ac.cn/grasscarp/ (accessed on 26 March 2014)). Sequence alignment of grass carp CRHs, UTS1, and CRHRs were conducted with ClustalX 2.1. The 3-D structure for grass carp CRHs, UTS1, and CRHRs were predicted and modeled by using the I-TASSER and SWISS-MODEL. To clarify the tissue distribution of grass carp CRHs, UTS1, and their receptors system, the total RNA was isolated from the olfactory bulb, telencephalon, optic tectum, cerebellum, medulla oblongata, hypothalamus, and pituitary, respectively. Then, specific primers were used for these target genes to test the expression level by real-time PCR (RT-PCR) and gel electrophoresis (Appendix A). In these studies, β-actin was used as an internal control [24].

### 2.4. Transfection and Luciferase Reporter Assay

For the functional expression of grass carp CRHR1 and CRHR2, the ORF of grass carp CRHR1 and CRHR2 were isolated by PCR and subcloned into the eukaryotic expression vector pcDNA3.1/zeo(-) to generate the CRHR1 (the vector name is pcDNA3.1/Zeo(-)-CRHR1) or CRHR2 (the vector name is pcDNA3.1/Zeo(-)-CRHR2) expression vector, respectively. The HEK293 cells were resuscitated and cultured to the appropriate density before seeding at a density of 0.05 × 10^6^ cells/0.5 mL/well in 24-well plates overnight. Next, transfection was carried out in 400 μL OPTI-MEM for 6 hr with 200 ng luciferase-expressing Luc reporter for the respective pathway, 10 ng pTK.RL (Promega, Madison, WI, USA), 20 ng pEGFP-N1 (Clontech, Mountain View, CA, USA), 10 ng pcDNA3.1/Zeo(-)-CRHR1 or pcDNA3.1/Zeo(-)-CRHR2, 90 ng pcDNA3.1, and 0.99 μL lipofectamine (2 mg/mL) with pBluescript II SK (Stratagene, La Jolla, CA, USA) as the carrier DNA to make up a total of 330 ng DNA for transfection. pTK.RL, the renilla luciferase-expressing reporter, was included to serve as an internal control, whereas the GFP-expressing vector pEGFP-N1 was used to monitor the transfection efficiency between the individual wells and separated experiments. Parallel transfection with the blank vector pcDNA3.1/Zeo without CRHR was used as the negative control. After transfection, the DMEM with 10% FBS solution was used to incubate the HEK293 cells for 18–24 h at 37 °C. Based on our validation, the duration of drug treatment has been optimized to 24 h for luciferin expression. After 24 h drug treatment, prepare ice-cold 1 × PBS to wash cells three times to remove the remaining drug and use lysis buffer (Promega) to dissolve the cells. Lysate samples of cells prepared were assayed for both firefly luciferase and renilla luciferase activities using a Dual-GloTM luciferase Assay Kit (Promega) in a Lumat LB9507 Luminometer (EG&G, Gaithburg, MD, USA). Transfection experiments were performed in quadruplicate with cells cultured in separate wells [24].

### 2.5. SLα and SLβ Secretion and mRNA Expression in Carp Pituitary Cells

Primary culture of grass carp pituitary cells was prepared by the trypsin/DNase digestion method as described previously [25]. Pituitary cells were obtained and seeded in poly D-lysine (Sigma, St. Louis, MO, USA) precoated 24-well cluster plates, with a density of 2.5 × 10^6^ cells/mL/well. The cells were incubated with test substances for 24 h. After that, the culture medium was harvested for monitoring the SLα and SLβ release. SLα and SLβ levels in these samples were quantified using ELISA with antibody raised against the respective hormones in carp species (for information on the respective antibody, please refer to Appendix A). In parallel experiments, total RNA was isolated from the pituitary cells, reverse-transcribed, and subjected to quantitative PCR for grass carp SLα and SLβ mRNA expression using a RoterGene-Q Real-Time PCR system (QIAGEN, Venlo, The Netherlands) (see Appendix A for primer sequences and PCR condition for the respective gene targets). In these studies, serial dilutions of plasmid DNA containing the ORF of SLα and SLβ cDNA were used as the standards for data calibration. Parallel real-time PCR measurement of β-actin was also conducted in an individual experiment to serve as the internal control [26].

### 2.6. Transcriptome Sequencing and Bioinformatics Analysis

Grass carp pituitary cells were evenly seeded in cell culture plates at 2.5 × 10^6^ cells/well/mL. After 16–18 h of culture, CRH1a was diluted with the testing medium to the desired working concentration. Four replicates each were set up for the control group and CRH1a treatment group. After 24 h of culture, the total RNA of pituitary cells was extracted by Trizol, the RNA concentration and purity were measured by the Nanodrop 2000 spectrophotometer, and the RIN value was detected by the Agilent 2100 bioanalyzer. For each 4 replicate groups, the samples were sent to Meiji Biomedical Technology Co., Ltd. (Tokyo, Japan) for sequencing, and the gene library was established. The overall quality of the RNA-seq was sufficient with an average of ~90% of the reads mapping to the grass carp genome. Then, the FPKM (fragments per kilobase million) value for each neuropeptide and their receptors were compared among the transcriptomic libraries. RSEM v1.2.7 was used to calculate the differential expression fold of different genes (FC). We used R Bioconductor Package, edgeR (4.0.2) to analyze the differentially expressed genes (DEGs) between different samples. Then, we analyzed the enrichment of differentially expressed genes. The grass carp hepatocytes were incubated with SLα and SLβ using the same method [27].

### 2.7. Immunofluorescence Staining of SLα and SLβ in Grass Carp Pituitary

To investigate the distribution of SLα and SLβ in grass carp, the pituitary of grass carp was isolated and fixed in 4% paraformaldehyde. The pituitary was sectioned under the condition of the maximum longitudinal section and a thickness of 4 mm. In the immunofluorescence experiment, the sections were first dewaxed and rehydrated. Citric acid antigen repair buffer was used to accomplish antigen retrieval. The sections were then incubated with 5% normal goat serum (Yeasen, Shanghai, China) for 1 h to block nonspecific binding. The primary antibodies against SLα (1:1000) and SLβ (1:1000) were diluted with 5% goat serum, and the sections were incubated overnight at 4 °C. The next day, 1 × PBS was used to wash the sections five times. The Alexa Fluor 594 Goat Anti-Rabbit IgG (1:200) (Yeasen) was diluted with 5% goat serum, and the sections were incubated for 1 h. Finally, the sections were counterstained with DAPI and mounted with an anti-fluorescence quenching medium. The stained sections were observed under a LSM710 confocal microscope (Carl Zeiss, Oberkochen, Germany) [28].

### 2.8. Single Cell Transcriptome Sequencing of Grass Carp Pituitary

An appropriate number of pituitary cells were combined with the reverse transcription reagent and subsequently introduced into the sample well within the SeekOne DD Chip S3 (Chip S3). Following this, barcoded hydrogel beads (BHBs) and partitioning oil were separately dispensed into their corresponding wells within Chip S3. Once emulsion droplets were generated, reverse transcription was executed at a temperature of 42 °C for a duration of 90 min, followed by inactivation at 85 °C for 5 min. Subsequently, cDNA was extracted from the broken droplets and subjected to amplification through PCR reactions. The resulting amplified cDNA product underwent a series of steps including purification, fragmentation, end repair, A-tailing, and ligation to sequencing adaptors. Finally, indexed PCR was performed to amplify the DNA, which represented the 3′ polyA region of expressing genes and also included the Cell Barcode and Unique Molecular Index. The indexed sequencing libraries were subjected to purification using SPRI beads, followed by quantification using quantitative PCR (KAPA Biosystems KK4824). The libraries were then sequenced either on an Illumina NovaSeq 6000 platform with PE150 read length or a DNBSEQ-T7 platform with PE150 read length [29].

### 2.9. Data Transformation and Statistical Analysis

For SLα and SLβ secretion measurement, standard curves with detectable range from 0.98 to 500 ng/mL were used for data calibration with the four-parameter logistic regression model of Prism 6.0 (GraphPad, San Diego, CA, USA). For real-time PCR of SLα and SLβ mRNA measurement, standard curves with a dynamic range of ≥105 and a correlation coefficient ≥ 0.95 were used for data calibration. Since no significant changes were noted for β-actin mRNA in our experiment, SLα and and SLβ mRNA data as well as the corresponding data were simply transformed as a percentage of the mean value in the control group without drug treatment (as “%Ctrl”). For individual samples of the luciferase reporter assay, firefly luciferase activity detected was routinely normalized against Renilla luciferase activity expressed in the same well and expressed as the luciferase activity ratio (as “Luc Ratio”). The data presented (as Mean ± SEM) were pooled results from 6–8 separate experiments and analyzed with ANOVA followed by Tukey’s test using Prism 6.0, and differences between groups were considered as significant at *p* < 0.05 (homogeneity of variance tests were performed).

## 3. Results

### 3.1. Sequence Analysis of CRHs and CRHRs in Grass Carp

The full-length of grass carp CRH1a (Appendix A), CRH1b (Appendix A), and UTS1 (Appendix A) were isolated from grass carp hypothalamus. Similar to other vertebrate models, grass carp CRH1a precursor also encoded a 41-a.a. CRH1a mature peptide (Appendix A), and CRHR1b precursor encoded a 41-a.a. CRH1b mature peptide (Appendix A) [2]. Sequence alignment showed that the mature peptides of CRH1a and CRH1b have 8 amino acid differences and CRH1a and UTS1 have 17 amino acid differences (Figure 1A). In addition to CRH, we have also isolated the full-length CRHR1 and CRHR2 from grass carp pituitary. Similar to other vertebrates, the deduced protein sequence also reveals that grass carp CRHR1 (Appendix A) and CRHR2 (Appendix A) both contain seven hydrophobic transmembrane domains (TMD1–7) linked with three extracellular (ECL1–3) and three intracellular loops (ICL1–3) (Figure 1C) [30]. Tissue distribution analysis showed that CRH1a and UTS1 were widely distributed in various brain regions of grass carp, such as telencephalon and hypothalamus, but CRH1b was lowly detected in the brain (Figure 1E). In this case of CRHR1 and CRHR2, transcript signals for CRHR1 could be detected at high levels in the hypothalamus and pituitary, and the transcript signals for CRHR2 were also mainly detected in various brain regions (Figure 1D).

To further characterize the ligand–receptor interactions and signal transduction pathways of carp CRHR1 and CRHR2, two lines of HEK-293 cells with stable expression of carp CRHR1 or CRHR2 were established, respectively, and used for transfection study with various luciferase-expression vectors. In the present study, HEK-293 cells with stable expression of CRHR1 or CRHR2 were transient transfected with cAMP response element (CRE)- or nuclear factor of activated T-cells (NFAT)-luciferase-expressing vectors to allow for the functional evaluation of the activation status of PKA- and Ca^2+^-dependent pathways, respectively. As shown in Figure 1C, for both CRHRs, CRH1a and UTS1 were more highly potent in cAMP-dependent signal pathways than in the Ca^2+^-dependent pathway. In addition, carp CRH1a and UTS1 were found to share similar activating potencies for the two CRHRs. Particularly, UTS1 has a stronger ability to activate the receptors (Figure 1E).

### 3.2. Transcriptomic Analysis for the Pituitary Actions of CRH1a in Grass Carp

To investigate the direct pituitary actions of CRH1a, primary cultured pituitary cells derived from prepuberty grass carp were incubated with CRH1a (1 μM). After 24 h treatment, high-throughput RNA-seq was utilized to compare the mRNA expression difference between the control and CRH1a treated groups. As a result, we detected 6109 differentially expressed genes (DEGs) (*p*-value < 0.05), which included 488 up-regulated (FC > 2) genes and 485 down-regulated (FC < 0.5) genes (Figure 2A). Up- and down-regulated DEGs were constructed according to the conditions (*p*-value < 0.05, FC > 1 or FC < 1) for KEGG enrichment analysis. To further understand the direct pituitary actions of CRH1a, annotated pathways of DEGs were analyzed using the KEGG database. The up-regulated DEGs were mostly enriched in ‘Endocrine system’, ‘Signal transduction’, ‘Nervous system lipid metabolism’, ‘Glutathione metabolism’, while the down-regulated DEGs were mainly enriched in ‘insulin resistance’, ‘prolactin signaling pathway’ (Figure 2B).

According to the KEGG analysis, several key genes have been identified in pathways related to endocrine system, signal transduction, immune system, and nervous system (Figure 2C). Among them, igf1r, insr, homer, and other genes related to signal transduction were significantly up-regulated, and the metabolism-related genes such as slβ were significantly up-regulated (Figure 2C). In addition, 24 h static incubation with CRH1a (1 μM) was able to significantly elevate the slβ mRNA expression, and the expression of slα and lhβ was slightly increased, but without significantly altering the gh, fshβ, cga, tshβ, prl, pomc1, and pomc2 transcript expression (Figure 2D).

### 3.3. CRH Could Induce Pituitary Somatolactin Secretion and mRNA Expression

Two somatolactin protein isoforms, namely, SLα and SLβ, were identified in grass carp pituitary. Immunofluorescence analysis revealed that SLα and SLβ were both detected in the neurointermediate lobe (NIL) of pituitary, but they did not colocalize in the same cells (Figure 3A). Single-cell transcriptomic analysis also showed that both SLα and SLβ were expressed in different cells (Figure 3B, Appendix A). These results suggest that CRH could activate CRHR1 to regulate the SLα and SLβ expression. To explore the regulation of CRH1a on the SLα and SLβ expression in grass carp pituitary, CRH1a was used to incubate grass carp pituitary cells in a time-dependent manner, and then, RT-qPCR and ELISA were used to detect SLα and SLβ mRNA and protein release, respectively. Firstly, CRH1a (1µM) was used to incubate grass carp pituitary cells for 3 h, 12 h, and 24 h, respectively. The results revealed that CRH1a could significantly promote the SLβ release and mRNA expression in a time-dependent manner (Figure 3C), but could only induce the SLα release and mRNA expression at 12 and 24 h (Figure 3C). Furthermore, immunofluorescence analysis showed that CRH1a could also significantly induce the SLα and SLβ synthesis in grass carp pituitary cells (Figure 3D, Appendix A).

### 3.4. Signal Transduction for the Regulation of SLα and SLβ by CRH1a

To elucidate the signal transduction for SLα and SLβ regulation by CRH1a, the possible involvement of the cAMP-dependent pathway was examined. Co-treatment with the AC inhibitor MDL12330A (20 µM) or PKA inhibitor H89 (20 µM) could block the CRH1a-induced SLα (Figure 4A) and SLβ (Figure 4B) secretion and mRNA expression, respectively. In the parallel experiments, the CRH1a-induced SLα (Figure 4C) and SLβ (Figure 4D) release and mRNA expression in grass carp pituitary cells were abrogated by simultaneous incubation with the PLC inactivator U73122 (10 µM), PKC inhibitor GF109203X (20 µM), or IP3 receptor blocker 2-APB (100 µM), respectively. Finally, the effects of CRH1a on intracellular Ca^2+^ levels in grass carp pituitary cells were also investigated. The results revealed that the CRH1a-induced SLα (Figure 4E) and SLβ (Figure 4F) secretion and mRNA expression were found to be abolished by co-treatment with the VSCC inhibitor nifedipine (10 µM), CaM antagonist calmidazolium (1 µM), or CaMK-II blocker KN62 (5 µM), respectively.

### 3.5. The Function of SLα and SLβ in Grass Carp Liver

To elucidate the function of SLα and SLβ in grass carp, we incubated the grass carp liver cells with SLα (20 nM) or SLβ (20 nM), respectively. After 24 h treatment, high-throughput RNA-seq was utilized to compare the mRNA expression difference between the control and SLα/SLβ treated groups. KEGG enrichment analysis suggests that SLα and SLβ played an important role in energy metabolism in grass carp liver. After treatment of SLα or SLβ, many metabolism-related pathways are affected, such as ‘fat digestion and absorption’, ‘cholesterol metabolism’, and ‘steroid synthesis’ pathways. Moreover, SLα down-regulated genes were involved in the digestion, absorption, transport, and metabolism of carbohydrate and significantly affected genes in the glucagon signaling pathway (Figure 5A). Particularly, SLβ-induced DEGs were involved in the cortisol synthesis and secretion, suggesting that SLβ should be involved in the stress process (Figure 5B). SLα and SLβ also have an effect on the synthesis of steroids. The expression level of vitamin D3 (VD3) metabolism-related gene, namely cyp24a1, has been reduced by SLα (Figure 5C) and SLβ (Figure 5D). The expression of cyp21a, which was involved in cortisol synthesis, was found to be 14-times higher in comparison to that in the control group (Figure 5D).

## 4. Discussion

The CRH family is essential for modulating physiological responses to stress, emotional behavior, and vertebrate anxiety [31,32,33]. As we know, CRH could induce pituitary ACTH release [10], which could stimulate the synthesis and release of glucocorticoids in the adrenal gland [2]. Somatolactin is a fish-specific hormone released from the neurointermediate lobe (NIL) of the posterior pituitary [11]. A previous study has reported that ACTH was also located in the pituitary NIL region. Interestingly, our present study found that CRH1a could also promote somatolactin release and mRNA expression in the teleost pituitary. In mammals, CRHR activation constitutes a major step in triggering post-receptor signaling by various CRHs mediated through the activation of adenylate cyclase (AC), resulting in the production of cAMP and the subsequent stimulation of protein kinase A (PKA) [34]. In the present study, the luciferase analysis showed that the CRH/CRHR system could highly activate the cAMP response element (CRE) reporter. In addition, pharmacological analysis showed that the cAMP-dependent pathway was involved in the CRH-induced pituitary somatolactin release and mRNA expression. Similarly, previous studies have showed that the CRH-induced ACTH secretion was also mediated by the AC/cAMP/PKA signal pathway [30]. In addition, somatolactin secretion and gene expression are known to be regulated by GnRH [35], PACAP [14], and NKB [24] through the AC/cAMP/PKA signal pathway in teleost pituitary cells.

Somatolactin is a fish-specific hormone that belongs to the prolactin and GH family. Recently, two isoforms of somatolactin, SLα and SLβ, have been identified in the fish pituitary, e.g., in zebrafish [12] and grass carp [14], and are suspected to have overlapping and yet distinct functions [15]. Previous studies showed that SL exerted its action via specific cell membrane receptors such as the somatolactin receptor (SLR) that belong to the cytokinine/hematopoietin receptor subfamily [36]. Transcripts for SLR were found in various tissues with the highest levels in liver and fat, supporting the notion that a major function of SL is the regulation of lipid metabolism [37]. To investigate the function of somatolactins in liver, recombinant SLα or SLβ proteins were used to incubate the primary cultured grass carp hepatocytes. Transcriptomic analysis showed that several differential expression genes were involved in the regulation of lipid metabolism, such as peroxisome proliferator-activated receptor-gamma coactivator-1alpha (PGC-1a) and vitamin D3 (VD3) metabolism-related gene (cyp24a1). PGC-1a is a member of the family of transcription coactivators that affect energy metabolism [38], which is involved in the lipid metabolism of liver cells, reducing the deposition of fat in the liver. Up-regulation of PGC-1α can promote fatty acid oxidation in muscle and liver and reduce triglyceride synthesis and serum triglyceride level [39]. This suggests that SL could promote fat metabolism by up-regulating pgc-1a. Transcriptomic analysis also showed that SL could inhibit cyp24a1 mRNA expression, which is the key metabolism enzyme for VD3. A recent study reported that VD3 signaling could promote fatty acid oxidation by inducing pgc-1a expression [40]. Fatty acid β-oxidation could provide energy for the stress response, which is a high energy consuming process. A recent study demonstrated that when the cyprinid species *Onychostoma marolepis* were faced with cold stress, lipolysis was stimulated along with enhanced fatty acid β-oxidation for energy, while fatty acid synthesis was suppressed in the early stage [41]. These results indicated that CRH could influence the lipid metabolism in the liver by regulating the synthesis and secretion of pituitary somatolactin and indirectly participate in the stress response.

In summary, our present study found that CRH could significantly induce pituitary SLα and SLβ secretion and synthesis. Pharmacological analysis found that the CRH-induced pituitary somatolactin expression was mediated by the AC/cAMP/PKA, PLC/IP3/PKC, and Ca^2+^/CaM/CaMK-II pathways. Finally, transcriptomic analysis showed that both SLα and SLβ should play an important role in the regulation of lipid metabolism in primary cultured hepatocytes. These results indicated that CRH is a novel stimulator of somatolactins in teleost pituitary cells, which could induce lipid metabolism to provide energy for the stress response.

## 5. Conclusions

In this work, we found that corticotropin-releasing hormone was a new novel stimulator of somatolactin in grass carp pituitary. CRH-induced pituitary somatolactin expression was mediated by the cAMP-dependent pathway, IP3 signal transduction pathway, and calcium signaling pathway. Further study of the somatolactin function in grass carp liver cells show that both SLα and SLβ could regulate the lipid metabolism. CRH was well known for its function in participating in stress processes. Based on our results, there may be another pathway for the CRH response to stress. Specifically, CRH, through inducing the synthesis of SL to impact lipid metabolism, may participate in stress.

## Figures and Tables

**Figure 1 cells-12-02770-f001:**
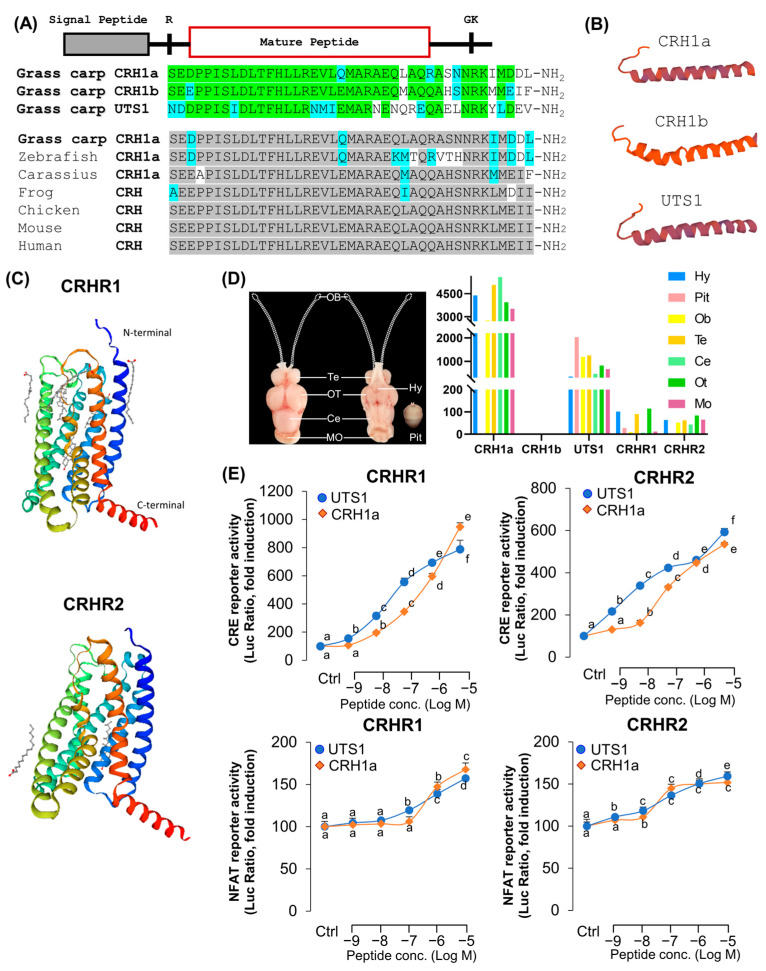
Structures of grass carp CRH and CRHRs. (**A**) Sequence alignment of grass carp mature peptides CRH1a, CRH1b, and UTS1. The same amino acid is marked in green and different amino acids are labeled with other colors. Sequence comparison of CRH among different vertebrate species. The same amino acid is marked in gray and different amino acids are labeled with other colors. (**B**) Three-dimensional protein structures of grass carp CRH1a, CRH1b, and UTS1. (**C**) Three-dimensional protein structures of grass carp CRHR1 and CRHR2. (**D**) Tissue distribution of CRHs and CRHRs in grass carp various brain areas (Hy: hypothalamus, Pit: pituitary, Ob: olfactory bulb, Te: olfactory bulb, Ce: cerebellum, Ot: optic tectum, Mo: medulla oblongata). Total RNA was extracted from various brain areas or pituitary of grass carp, and RT-PCR was performed by using specific primers. (**E**) The selectivity of CRH1a and UTS1 for receptor ligands, the HEK293T cells which expressed grass carp CRHR1 and CRHR2 were treated with various doses of CRH1a and UTS1, respectively. The data of different treatment groups were tested by Tukey, and the groups with significant differences were marked with different letters.

**Figure 2 cells-12-02770-f002:**
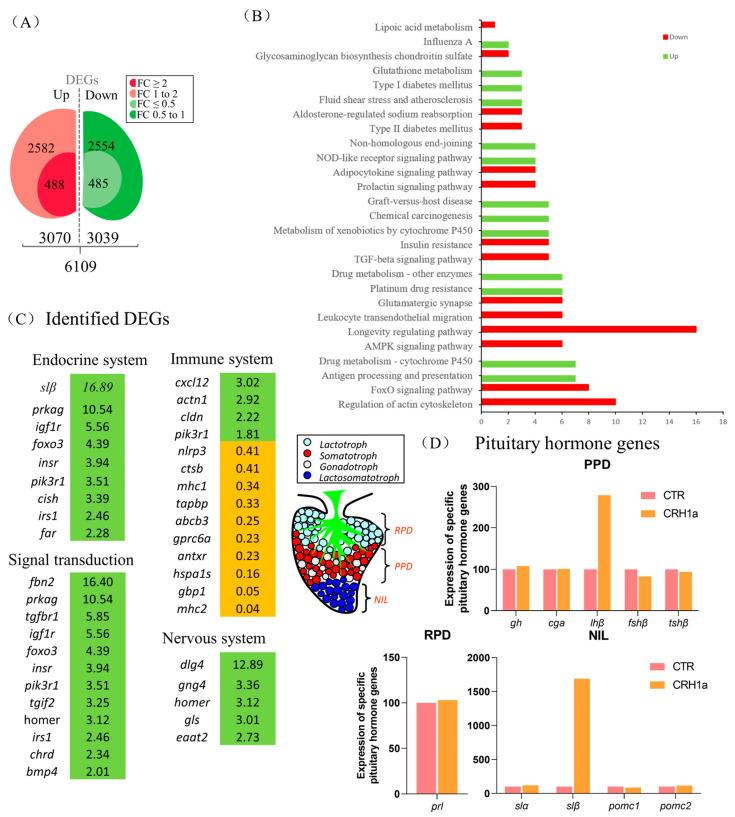
Transcriptomic analysis for the pituitary actions of CRH1a in grass carp. Kyoto Encyclopedia of Genes and Genomes (KEGG) analysis based on the transcriptome. (**A**) Differentially expressed genes were screened according to the conditions of *p*-adjust < 0.05, FC ≥ 2, 1 < FC < 2, FC ≤ 0.5, and 0.5 < FC < 1. (**B**) A total of 973 DEGs were classified into the different enriched pathways using KEGG analysis which included up-regulation and down-regulation DEGs. (**C**) The screened differentially expressed genes were classified according to the partial KEGG pathway. The increased expression of up-regulated genes was marked in green and orange in contrast. (**D**) Differential expression of specific pituitary hormone genes after CRH1a treatment based on the transcriptome (RPD: rostral pars distalis, PPD: proximal pars distalis, NIL: neurointermediate iobe).

**Figure 3 cells-12-02770-f003:**
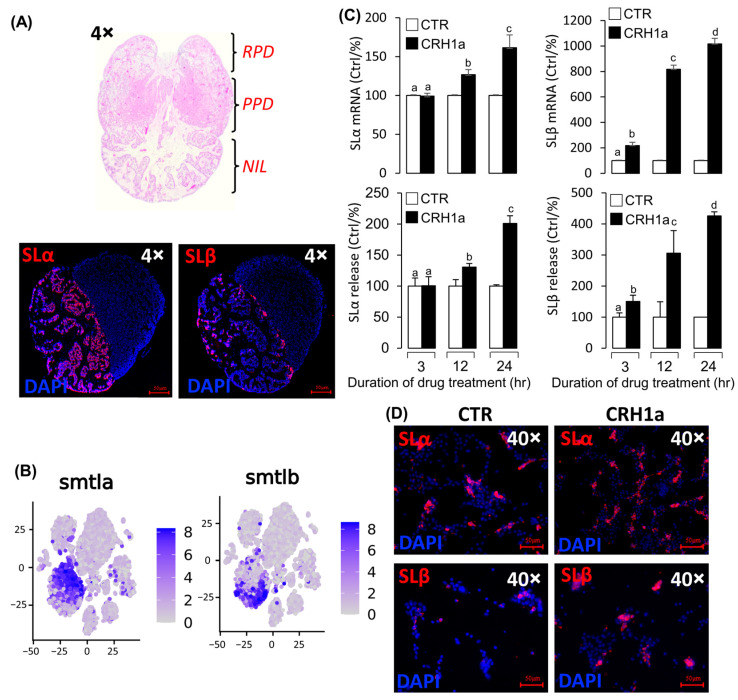
Pituitary SLα and SLβ regulation by CRH1a. (**A**) Hematoxylin–eosin (HE) staining results of pituitary tissue section and immunofluorescence localization of SLα/SLβ in grass carp pituitary tissue. (**B**) Single cell RNA sequencing (scRNA-seq) analysis of SLα/SLβ in grass carp pituitary. (**C**) Time course of grass carp CRH1a (1 μM) treatment on SLα/SLβ release and mRNA expression in grass carp pituitary cells. The data of different treatment groups were tested by Tukey test, and the groups with significant differences were marked with different letters. (**D**) The results of changes in SLα and SLβ protein levels using immunofluorescence after treatment of grass carp pituitary cells in vitro.

**Figure 4 cells-12-02770-f004:**
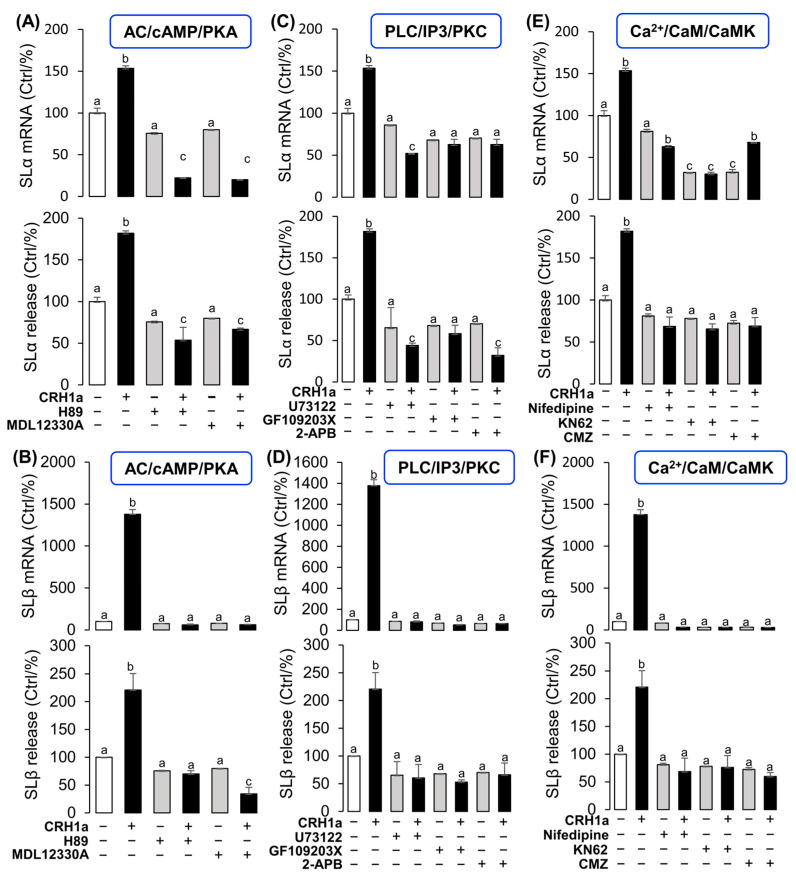
Signal transduction for the regulation of SLα and SLβ by CRH1a. Effects of 24-hr co-treatment with the AC inhibitor MDL12330A or PKA inhibitor H89 on CRH1a-induced (**A**) SLα and (**B**) SLβ mRNA expression and protein release. Effects of co-treatment with the PLC inactivator U73122, PKC inhibitor GF109203X or IP3 receptor blocker 2-APB on CRH1a-induced (**C**) SLα and (**D**) SLβ mRNA expression and protein release. Effects of co-treatment with VSCC inhibitor nifedipine, CaM antagonist calmidazolium or CaMK-II blocker KN62 on CRH1a-induced (**E**) SLα and (**F**) SLβ mRNA expression and protein release. After drug treatment, the total RNA of cells was extracted by Trizol method, and the expression of various genes was detected by RT-PCR. Data are presented as mean ± SEM and differences between groups are significant at *p* < 0.05 by marking different letters by using Tukey test. The inhibitors used in the experiment: MDL12330A (20 μM), H89 (20 μM), U73122 (10 μM), GF109203X (20 μM), 2-APB (100 μM), nifedipine (10 μM), Calmidazolium (1 μM), KN62 (5 μM).

**Figure 5 cells-12-02770-f005:**
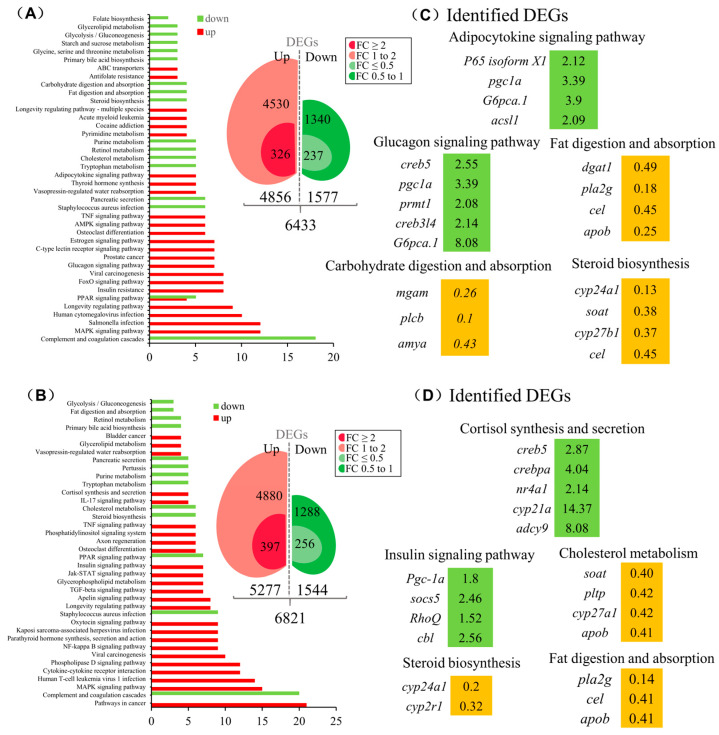
The function of SLα and SLβ in grass carp liver. KEGG analysis based on the transcriptome of SLα- (**A**) or SLβ- (**B**) induced DEGs. SLα- (**C**) or SLβ- (**D**) induced key DEGs were classified according to the partial KEGG pathway. The increased expression of up-regulated genes was marked in green and orange in contrast.

## Data Availability

The original contributions presented in the study are included in the article/Appendix A. Further inquiries can be directed to the corresponding author.

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
