# Peer review of "Corticotropin-Releasing Hormone: A Novel Stimulator of Somatolactin in Teleost Pituitary Cells"

_cells, 2023, doi:10.3390/cells12242770_

Round 1
Reviewer 1 Report
Comments and Suggestions for Authors
In this paper, Du et al. report that CRH is a novel stimulator of SL and through this, may participate in the stress response by regulating energy metabolism.
The study has been well conducted with various cutting-edge methodological approaches. Globally the results presented in this paper are convincing but needed to be presented in more details. Moreover, for some expriments (see below) additional controls are also needed.
Line 28 : “CRH was first isolated …”.
Line 29 : ” [2]. It is known for …”.
Lines 34-5: In teleosts, peptides of the CRH family were invlolved…”.
Line 36 : what does it mean : “000” ?
“… hypothalamus-pituitary-internal (HPI) axis”.
Line 37 : please add a reference demonstrating that « CRH could also promote fish pituitary ACTH release ».
Lines 42 and 347-8 : “… and are suspected to to have both overlapping and distinct functions”.
Line 47 : “in the rainbow trout (Onchorynchus mykiss), …”.
Line 49 : “In the female thinlip mullet (Liza ramada), …”..
Line 58 : “… we demonstrated…”.
Line 68 : what does it means : “+1” ?
Line 71 : “… then spinosectomized, according to …”.
Lines 82 and 126 : “… and reverse-transcribed …”.
Line 84 : “… specific primers designed from the putative sequences …”. Please specify from which database the gene sequences were found.
Line 98 : the vector name is pcDNA3.1/Zeo(-)-CRHR1.
Line 99 : 0.05x106
Lines 108-9 and 131 : “… as negative control…”.
Lines 120 ad 134: 2.5x106
Line 150 : “… functional region..”. What do the authors mean with these words ?
Line 151 : “… the pituitary of grass carp was isolated, obtained, and fixed.”. The word « obtained » is not appropriate. Please correct.
Lines 197 and 202-3: “… similar to other vertebrates …”. Please add a reference
Line 230 : “… were shown to share similar activating potency… “. could you elaborate on that please. I do not agree with your comment regarding the effects of UTS1 and CRH1a on the cAMP pathway, since UTS1 appears clearly more potent than CRH1a, particularly on CRHR2.
Line 236 : “As a result, we detected …”.
Line 239 : FC<1
Line 243 : please replace « together with » by « while ».
Line 258-9 : could you elaborate on that please. The elevation is very strong for slβ but much weaker for lhβ and slα.
Line 261 : please specify how the two SL isoforms were isolated. It is unclear whether the authors refer to the proteins or their cDNAs.
Line 265: “… showed that SLα and SLβ were mainly expressed in different cells”. The analysis of the scRNA seq results is too superficial. More information needs to be provided, particularly regarding the different cell clusters displayed.
Lines 265-6: “These results suggest that…”
Lines 273-4: “… immunofluorescence analysis showed that CRH1a could also significantly induce SLα and SLβ synthesis”. I disagree with this sentence, because the study does not provide any quantification of the labeled cells. Please provide accurate data regarding the number of labeled cells.
Lines 309-10: “KEGG enrichment analysis suggests that SLα and SLβ played an important role in energy metabolism in grass carp liver”.
Line 312: “… steroid synthesis… ”.
Line 316: “… secretion, suggesting that SLβ is involved in…”.
Line 318: “ SLα and SLβ also have an effect on …”.
Lines 319-20: “The expression levels of the vitamin D3 metabolism-related gene, namely cyp24a1, has been reduced by SLα …”.
Lines 330-1: please add a reference.
Lines 332-3: “… CRH1a could also promote somatolactin release and mRNA expression in the teleost pituitary”.
Line 335: “ … resulting in production of cAMP and subsequent stimulation of …”.
Lines 355-6: “… vitamin D3 metabolism-related gene (cyp24a1) …”.
Line 362: please specify the abbreviation “VD3”.
Line 364: “Fatty acid β-oxidation could provide energy for stress response, which is a high energy consuming process.”.
Line 365: “… when the cyprinid species Onychostoma marolepsis is faced with …”.
Line 376-7 “These results indicated that CRH is a novel stimulator…”.
Fig.1 and its legend : please replace « cloning » by « structure » ; (A) Please change the text as follows: “Sequence alignment of grass carp mature peptides CRH1a, CRH1b and UTS1 and sequence comparison of CRH among different vertebrate species”. (B) and (D) please suppress « The ». (E) Please replace the table by a graph. Moreover, please specify in what units the values are expressed.
All figures must be numbered following their number of appearance : but in the text, Fig.1C is described after Fig. 1E.
Fig. 2 and its legend : (C) According to Fig. 2D, I suppose that « sl » is « slβ ». If so, please correct. (D) Please replace the table by a graph and specify the meaning of RPD, PPD and NIL.
Fig. 3 and its legend : (A) negative control are missing. They must be added in order to prove that the antibodies used did not recognize inappropriate material. (D) Please specify the meaning of the values. The different cell clusters displayed in the figure should be specified.
Once again, all figures must be numbered following their number of appearance : but in the text, Fig.3D is described after Fig. 1C.
Fig. 5 and its legend. Once again, all figures must be numbered following their number of appearance : but in the text, Fig.5B is described after Fig. 5C.
In Fig. 2C, up-regulated genes are in yellow and down-regulated genes are in green. But the opposite applies in Fig.5B and 5D. Why ? Please homogenize.
Comments on the Quality of English Language
I advise the authors to find a native English speaker to proofread the manuscript.
Author Response
Responses to Reviewer 2#:
Comments and Suggestions for Authors
Evaluation report of the Ms entitled “Corticotropin-Releasing Hormone: a Novel Stimulator of Somatolactin in Teleost Pituitary Cells” from Du et al.
General evaluation: This is an original and innovative work in fish physiology, using biochemical and molecular tools to test an important hypothesis. The introduction is competent, citing appropriate works, with a clear rational and detailed objectives. Material and methods section is well detailed, but I assume that some of the laboratorial protocols followed pre-existent works that are not cited. Moreover, authors mention a pairwise test (Dunnett) following a One-Way ANOVA that is not appropriate for this dataset. Results are clear with nice pictures, schemes and tables. But attention; all figure captions should be self-explanatory. Discussion flows well, well-supported by the results and appropriate references. But the last paragraph (conclusion) should include some reference to future works. For more details, see Specific evaluation.
Specific evaluation:
1.L22-23: As general scientific style rule, s. Replace “somatolactin” and “corticotropin-releasing hormone” by related one’s words.
Response:
Thank you very much for your comments. We have replaced the key words.
Keywords: pituitary; neurointermediate lobe; stress; signal transduction; lipid metabolism Line 22
2.L35: “regulation of fish osmotic pressure” is better to replace by “osmoregulation”
Response:
Thank you very much for your comments. We have amended this description.
In teleost, peptides of CRH family were involved in the fish osmoregulation, glucocorticoid release (stress), food intake, and body exercise, through the hypothalamus-pituitary- internal (HPI) axis [6, 7, 8, 9]. Line 33-35
3.L36: “000” !?
Response:
Thank you very much for your comments. We have amended this error.
In teleost, peptides of CRH family were involved in the fish osmoregulation, glucocorticoid release (stress), food intake, and body exercise, through the hypothalamus-pituitary- internal (HPI) axis [6, 7, 8, 9]. Line 33-35
4.L47-48: Put in italics the rainbow trout scientific species name.
Response:
Thank you very much for your comments. We have amended this error.
In rainbow trout (Oncorhynchus mykiss), the environmental stress could cause rapid activation of SL-secreting cells [20] Line 47
5.L55: “grass carp” also use the scientific name
Response:
Thank you very much for your comments. We have amended this error.
In the present study, the correlation between CRH and somatolactins and their functions in stress were examined in grass carp (Ctenopharyngodon idellus), a commercial fish in Asian countries with high market value [23]. Line 55
6.L55-56: “a commercial fish in Asian countries with high market value”. Seems unnecessary. But to keep kit you need to add a reference to support it
Response:
Thank you very much for your comments. The reference has been supplemented.
In the present study, the correlation between CRH and somatolactins and their functions in stress were examined in grass carp (Ctenopharyngodon idellus), a commercial fish in Asian countries with high market value [23]. Line 55
Lin SY, Marco M, Gao YL, Wong MH. Sustainable management of non-native grass carp as a protein source, weed-control agent and sport fish. Aquaculture Research, 2022, 53, 5809–5824.
7.L58-59: “we have demonstrated for the first time that CRH could significantly induce pituitary SLα and SLβ secretion and mRNA expression”. Remove it or transform. This is the objectives section
Response:
Thank you very much for your comments. We have deleted it. Line 55-57
8.L68: Put in italics the grass carp scientific species name.
Response:
Thank you very much for your comments. We have amended this error.
One-year-old grass carp (Ctenopharyngodon idellus) with a body weight of 1.0 to 1.5 kg and a body length of 40 to 50 cm were acquired from local markets and maintained one week in well-aerated 250-L aquaria under a 14-hour light/10-hour dark photoperiod at 20±2°C. Line 66
9.L68-69: “1.5 to 1.0 kg” invert order please. Add length if possible.
Response:
Thank you very much for your comments. We have amended this error and added the length of grass carp.
One-year-old grass carp (Ctenopharyngodon idellus) with a body weight of 1.0 to 1.5 kg and a body length of 40 to 50 cm were acquired from local markets and maintained one week in well-aerated 250-L aquaria under a 14-hour light/10-hour dark photoperiod at 20±2°C. Line 66-68
10.L66: I would enjoy better 2.1 Fish acquisition, acclimation and sacrifice. Section 2.2. could be the Chemical reagents
Response:
Thank you very much for your comments. We have changed the titles.
2.1. Fish acquisition, acclimation and sacrifice Line 65
2.2. Chemical reagents Line 73
11.L69: “maintained” how many time? Two-weeks? Less?
Response:
Thank you very much for your comments. We have added the length of grass carp.
One-year-old grass carp (Ctenopharyngodon idellus) with a body weight of 1.0 to 1.5 kg and a body length of 40 to 50 cm were acquired from local markets and maintained one week in well-aerated 250-L aquaria under a 14-hour light/10-hour dark photoperiod at 20±2°C. Line 66-68
12.L81-93: No work is cited to support this methodological approach. The same for sections 2.3, 2.5, 2.6 and 2.7
Response:
Thank you very much for your comments. The reference has been supplemented. Reference 24-29
Hu GF, He ML, Ko WK, Lin CY, Wong AO. Novel pituitary actions of TAC3 gene products in fish model: -Receptor specificity and signal transduction for prolactin and somatolactin alpha regulation by neurokinin B (NKB) and NKB-related peptide in carp pituitary cells. Endocrinology, 2014, 155(9): 3582-3596.
Hu Q, Qin Q, Xu S, Zhou L, Xia C, Shi X, Zhang H, Jia J, Ye C, Yin Z, Hu G. Pituitary Actions of EGF on Gonadotropins, Growth Hormone, Prolactin and Somatolactins in Grass Carp. Biology, 2020, 9(9), 279.
Xia C, Qin X, Zhou L, Shi X, Cai T, Xie Y, Li W, Du R, OuYang Y, Yin Z, Hu G. Reproductive Regulation of PrRPs in Teleost: The Link Between Feeding and Reproduction.Frontiers in endocrinology, 2021, 12, 762826.
Im K, Mareninov S, Diaz MFP, Yong WH. An Introduction to Performing Immunofluorescence Staining. Methods in molecular biology (Clifton, N.J.), 2019, 1897, 299–311.
Siddique K, Ager-Wick E, Fontaine R, Weltzien FA, Henkel CV. Characterization of hormone-producing cell types in the teleost pituitary gland using single-cell RNA-seq. Scientific data, 2021, 8(1), 279.
13.L161: Please also add the magnification
Response:
Thank you very much for your comments. We have added the magnification. Line 283
14.L191: Did you check the parametric test prerequisites? Normality and homogeneity of variances?
Response:
Thank you very much for your comments. We have checked the parametric test prerequisites and. homogeneity of variance tests were performed before the Tukey test.
The data presented (as Mean ± SEM) were pooled results from 6-8 separate experiments and analyzed with ANOVA followed by Tukey’s test using Prism 6.0 and differences between groups were considered as significant at P<0.05 (homogeneity of variance tests were performed). Line 191-194
15.L191: You cannot use Dunnett test to see differences between groups!? This is used when you are comparing each treatment with a control. You should use Tukey or similar.
Response:
Thank you very much for your comments. We have correct the test method.
The data presented (as Mean ± SEM) were pooled results from 6-8 separate experiments and analyzed with ANOVA followed by Tukey’s test using Prism 6.0 and differences between groups were considered as significant at P<0.05 (homogeneity of variance tests were performed). Line 191-194
16.L213-220: Figure captions should be self-explanatory (Hy, Pit, …, ?)
Response:
Thank you very much for your comments. The full name was added.
Figure 1C: Three-dimensional protein structure of grass carp CRHR1 and CRHR2. (D) Tissue distribution of CRHs and CRHRs in grass carp various brain areas (Hy: hypothalamus, Pit: pituitary, Ob: olfactory bulb, Te: olfactory bulb, Ce: cerebellum, Ot: optic tectum, Mo: medulla oblongata.). Total RNA was extracted from various brain areas or pituitary of grass carp, and RT-PCR was performed by using specific primers. Line 219-220
17.L277-282: Figure captions should be self-explanatory (RPD, PPd and NIL?)
Response:
Thank you very much for your comments. The full name was added.
Figure 2D: Differential expression of specific pituitary hormone genes after CRH1a treatment based on the transcriptome (RPD: rostral pars distalis, PPD: proximal pars distalis, NIL: neurointermedi-ate iobe). Line 256-257
18.L298-304: Are you using a Tukey (different letters) or a Dunnett (we use *) test?
Need also to say that each scale bar represents mean and SEM.
Response:
Thank you very much for your comments. The method used in this study is Tukey test and the data presented as Mean ± SEM
The data presented (as Mean ± SEM) were pooled results from 6-8 separate experiments and analyzed with ANOVA followed by Tukey’s test using Prism 6.0 and differences between groups were considered as significant at P<0.05 (homogeneity of variance tests were performed). Line 191-194
19.L327: “and so on” !?
Response:
Thank you very much for your comments. We have amended this error.
The CRH family is essential for modulating physiological responses to stress, emotional behavior, vertebrate anxiety [31, 32, 33]. Line 333-334
20.L330: “A previous study….”
Response:
Thank you very much for your comments. We have amended this error.
A previous study has reported that ACTH was also located in the pituitary NIL region. Line 337-338
21.L344: Add a reference to support it.
Response:
Thank you very much for your comments. We have amended this error.
In addition, somatolactin secretion and gene expression are known to be regulated by GnRH [35], PACAP [14] and NKB [24] through AC/cAMP/PKA signal pathway in teleost pituitary cells. Line 347-350
Hillhouse EW, Grammatopoulos DK. The molecular mechanisms underlying the regulation of the biological activity of corticotropin-releasing hormone receptors: implications for physiology and pathophysiology. Endocr Rev, 2006, 27(3): 260-286.
22.L362: “A recent study…”
Response:
Thank you very much for your comments. We have amended this error.
A recent study reported that VD3 signaling could promote fatty acid oxidation by inducing pgc-1a expression [40] Line 369-370
23.L365: italics for the species name
Response:
Thank you very much for your comments. We have amended this error.
A recent study demonstrated that when the cyprinid species Onychostoma marolepis were faced with cold stress, the lipolysis was stimulated along with the enhanced fatty acid β-oxidation for energy, while the fatty acid synthesis was suppressed in the early stage [41] Line 372-373
Comments on the Quality of English Language
A very few minor corrections.
Response:
Thank you very much for your comments. We have amended language errors in the text.

Reviewer 2 Report
Comments and Suggestions for Authors
Evaluation report of the Ms entitled “Corticotropin-Releasing Hormone: a Novel Stimulator of Somatolactin in Teleost Pituitary Cells” from Du et al.
General evaluation: This is an original and innovative work in fish physiology, using biochemical and molecular tools to test an important hypothesis. The introduction is competent, citing appropriate works, with a clear rational and detailed objectives. Material and methods section is well detailed, but I assume that some of the laboratorial protocols followed pre-existent works that are not cited. Moreover, authors mention a pairwise test (Dunnett) following a One-Way ANOVA that is not appropriate for this dataset. Results are clear with nice pictures, schemes and tables. But attention; all figure captions should be self-explanatory. Discussion flows well, well-supported by the results and appropriate references. But the last paragraph (conclusion) should include some reference to future works. For more details, see Specific evaluation.
Specific evaluation:
L22-23: As general scientific style rule, keywords should not repeat words already existent in the title. Replace “somatolactin” and “corticotropin-releasing hormone” by related one’s words.
L35: “regulation of fish osmotic pressure” is better to replace by “osmoregulation”
L36: “000” !?
L47-48: Put in italics the rainbow trout scientific species name.
L55: “grass carp” also use the scientific name
L55-56: “a commercial fish in Asian countries with high market value”. Seems unnecessary. But to keep kit you need to add a reference to support it
L58-59: “we have demonstrated for the first time that CRH could significantly induce pituitary SLα and SLβ secretion and mRNA expression”. Remove it or transform. This is the objectives section
L68: Put in italics the grass carp scientific species name.
L68-69: “1.5 to 1.0 kg” invert order please. Add length if possible.
L66: I would enjoy better 2.1 Fish acquisition, acclimation and sacrifice. Section 2.2. could be the Chemical reagents
L69: “maintained” how many time? Two-weeks? Less?
L81-93: No work is cited to support this methodological approach. The same for sections 2.3, 2.5, 2.6 and 2.7
L161: Please also add the magnification
L191: Did you check the parametric test prerequisites? Normality and homogeneity of variances?
L191: You cannot use Dunnett test to see differences between groups!? This is used when you are comparing each treatment with a control. You should use Tukey or similar.
L213-220: Figure captions should be self-explanatory (Hy, Pit, …, ?)
L277-282: Figure captions should be self-explanatory (RPD, PPd and NIL?)
L298-304: Are you using a Tukey (different letters) or a Dunnett (we use *) test? Need also to say that each scale bar represents mean and SEM.
L327: “and so on” !?
L330: “A previous study….”
L344: Add a reference to support it.
L362: “A recent study…”
L365: italics for the species name
Comments on the Quality of English LanguageA very few minor corrections.
Author Response

(The authors gave the same response as above.)

Round 2
Reviewer 1 Report
Comments and Suggestions for Authors
The authors have revised their manuscript appropriately. I only recommand the following minor changes.
Lines 36-37: hypothalamus-pituitary-interrenal axis (HPI).
Interrenal and not internal.
Line 71: followed by spinosectomy
I apologize to the authors for the mistakes made in my previous revision. Lines 121-123: this sentence is poorly written. Please correct. Lines 288-90: Immunofluorescence analysis revealed that SLα and SLβ were both detected in neurointermediate lobe (NIL) of pituitary but they did not colocalized in the same cells (Figure 3A). Lines 250-252: In addition, carp CRH1a and UTS1 were found to share similar activating potent for two CRHRs (Figure 1C). Please suppress "be".
Author Response
Comments and Suggestions for Authors
The authors have revised their manuscript appropriately. I only recommand the following minor changes.
1.Lines 36-37: hypothalamus-pituitary-interrenal axis (HPI). Interrenal and not internal.
Response:
Thank you very much for your comments. We have amended this error.
In teleost, peptides of CRH family were involved in the fish osmoregulation, glucocorticoid release (stress), food intake, and body exercise, through the hypothalamus-pituitary- interrenal (HPI) axis [6, 7, 8, 9]. Line 33-35
2.Line 71: followed by spinosectomy
Response:
Thank you very much for your comments. We have amended this error.
During the process, the fish were anesthetized in 0.05% MS222 (Sigma, St. Louis, MO) followed by spinosectomy, according to the protocol approved by the committee for animal use at Huazhong Agricultural University (Ethical Approval No. HBAC20091138; Date: 15 November 2009). Line 70-72
I apologize to the authors for the mistakes made in my previous revision
3.Lines 121-123: this sentence is poorly written. Please correct.
Response:
Thank you very much for your comments. We have amended this sentence.
Pituitary cells were obtained and seeded in poly D-lysine (Sigma, St. Louis, MO) precoated 24-well cluster plates, with a density of 2.5 × 106 cells/mL/well. The cells were incubated with test substances for 24 h. Lines 121-123
4.Lines 288-90: Immunofluorescence analysis revealed that SLα and SLβ were both detected in neurointermediate lobe (NIL) of pituitary but they did not colocalized in the same cells (Figure 3A).
Response:
Thank you very much for your comments. We have amended the description of result.
Immunofluorescence analysis revealed that SLα and SLβ were both detected in neurointermediate lobe (NIL) of pituitary, but they did not colocalized in the same cells
Line 269-270
5.Lines 250-252: In addition, carp CRH1a and UTS1 were found to share similar activating potent for two CRHRs (Figure 1C). Please suppress "be".
Response:
Thank you very much for your comments. We have amended this error.
In addition, carp CRH1a and UTS1 were found to share similar activating potent for two CRHRs. Particularly, UTS1 have a stronger ability to activate the receptors. (Figure 1E). Line 237-238
